# Defining Targets for Reversing Declines of Soil Carbon in High-Intensity Arable Cropping

**Geoffrey R. Squire \***[ID]**, Mark Young, Linda Ford, Gillian Banks and Cathy Hawes**

James Hutton Institute, Invergowrie, Dundee DD2 5DA, Scotland, UK; mark.young@hutton.ac.uk (M.Y.); linda.ford@hutton.ac.uk (L.F.); gill.banks@hutton.ac.uk (G.B.); cathy.hawes@hutton.ac.uk (C.H.)

**\*** Correspondence: geoff.squire@hutton.ac.uk

**Abstract:** Soil organic carbon (SOC) is declining globally due to intensification of agriculture. Reversing declines should reduce soil erosion, maintain yields, raise the soil's atmospheric carbon sink, and improve habitat for biodiversity. Commercial fields were sampled in a diverse European Atlantic zone cropland to relate SOC status to cropping intensity and to define a realistic target for restoration. SOC (%C by mass) decreased from 4% to 2% as the proportion of high-intensity crops increased from zero to 55% (linear regression, F pr. < 0.001). In further sampling in and around high-intensity fields, mean SOC increased from 2.4% in cultivated soil to 3.3% in field margins and 4.8% in nearby uncultivated land (F pr. < 0.001). Three broad zones of SOC in close spatial proximity were then defined: 1) high-intensity arable from 1% to 3%, 2) mid-intensity arable and arable-grass from 3% to 5% and 3) uncultivated and semi-natural land from 5% upwards. C:N ratio was constrained around 12, unaffected by cropping intensity, but slightly lower in fields than in margins and uncultivated land (F pr. < 0.001). A feasible target SOC of just above 3% was defined for high-intensity sites. There should be no biophysical obstacle to raising SOC above 3% in the high-input sector. Results argue against treating cropland of this type as uniform: assessment and restoration should be implemented field by field.

**Keywords:** carbon; nitrogen; C:N ratio; soil organic carbon; SOC; intensified agriculture; arable; pasture; cereal; grass

## 1. Introduction

The organic carbon status of soil and the stoichiometry between carbon (C) and other elements, particularly nitrogen (N) and phosphorus (P), are increasingly viewed as broad-scale indicators of agro-ecosystem functioning and health [1–3]. Notably, a loss of soil organic carbon (SOC) during the intensification of agricultural land leads to reduced soil cohesion, an increased risk of erosion, degradation of soil properties essential for plant growth and increased greenhouse gas emissions, notable carbon dioxide and nitrous oxide [1,4]. The pre-intensification levels of SOC and the rates at which it is lost or gained depend on other major factors including climate and soil type, which themselves influence the type and intensity of agriculture that is possible in a region. Over the earth's agricultural surface, therefore, SOC varies over a very wide range. In contrast, the stoichiometry of carbon and other major elements, particularly N, is more constrained in response to intensification than SOC itself [2] and has in some cases provided a subtle indicator of the effects of crop and pasture management on soil [5–7].

Despite underlying complexity, substantive progress has been made in estimating the carbon-balance, soil carbon stores and carbon stoichiometry in agriculture. Modelling platforms are now capable of predicting carbon stores and fluxes over extensive tracts of land [8–10] and defining the most effective options for restoration [11,12]. However, a limitation in many cases is a shortage of data on soil characteristics and the associated climate and land management. Models rely first for

construction and then testing and corroboration on adequate ground survey and farmland inventory. In this respect, a major question being widely considered in carbon modelling and soil science generally is the degree of site-specificity and spatial resolution needed to predict soil status and define options for restoration. For example, the modelling of SOC at sites in the wheat belt of east Australia points to the need for detail on agricultural practices, land use history and local environment, tailored to specific agro-ecological scales [8]. Such deliberations are part of broader questions being asked of the type, relevance and resolution of data on soils that would be most useful in management and restoration [13,14].

Many of these issues on scale and spatial resolution have been raised in European studies that aim to characterise agricultural land in relation to farming intensity. For example, a land-class named High Nature Value Farmland or HNVF [15–17], which is recognised in the European Union (EU) as supporting biodiversity and crucial ecological functions, was found to be associated with higher soil carbon content than conventional high-input farmland [18]. The association was based on the Land Use/Land Cover Area Frame Survey, or LUCAS, [19], which compiles topsoil data from point surveys. The spatial scale of the original HNVF data with the LUCAS points overlaid on it [18] may be adequate for defining broad biogeographical zonation, but within any designated parcel of land, there could be many different types of land use and levels of SOC [17]. The need for site-specific data would be essential where the risk from intensely managed agriculture might be undetected in broad-scale mapping.

A long-term regional study area in eastern Scotland, UK, offers the potential to explore such questions. It contains a wide range of farming intensity and non-farmed land in close proximity [20]. On the map of HNVF in Europe [16,18], most of the area in question is assigned to conventional intensified farmland, yet the area is far from uniform. Agriculture went through a trajectory of intensification after the mid-1900s, in which depth of cultivation, mineral fertiliser and pesticide inputs all increased, but did not lead to a single form of cropping [20,21]. Rather, a variety of markets and farming preferences resulted in a varied agriculture, comprising a mosaic of conventional farming, HNVF and semi-natural land, among which the most intensified fields may be associated with impaired soil properties [22,23] and high erosion risk [24]. Present understanding of SOC status in arable land is limited by a low incidence of sample points in soil survey grids [25] and a restricted geographical distribution of long-term study sites available for resampling after intensification [26].

Nevertheless, the proximity of different land use types in the region should enable comparison of SOC over a wide range of farming intensity in the same broad climatic and farming conditions. The main aims of this study were to characterise the association between SOC and cropping intensity in commercial fields; to compare SOC in fields and in nearby uncultivated land as a means of quantifying decline due to repeated cultivation; and to identify targets for rehabilitation, in terms of field-types most at risk and achievable targets for SOC. Options for rehabilitation of soil are considered.

## 2. Materials and Methods

### 2.1. Region, Sites and Sampling

The region of arable-grass cropland in the east of Scotland has been described previously in terms of cropping systems and agronomic inputs [20,21]. The land is in productive, temperate agriculture, in a oceanic climate within 35 km from the coast, over a north-south extent of 250 km around latitude 56N, to the east or right of the zig-zag line on the map in Figure 1a. Soils have been well characterised [27]. Maps and data for World Reference Base classes and other soil criteria are available online [28]. The land supports a range of land use including long term pasture and a mix of arable crops and grass in varying sequences and combinations. Cropping systems in the arable-grass sector of decreasing intensity were defined from high-input potato and cereals (system VI), winter cereals (V), mixed cereals (IV), spring cereals (III), spring cereals alternating with grass (II) and grass with occasional arable (I) [21].

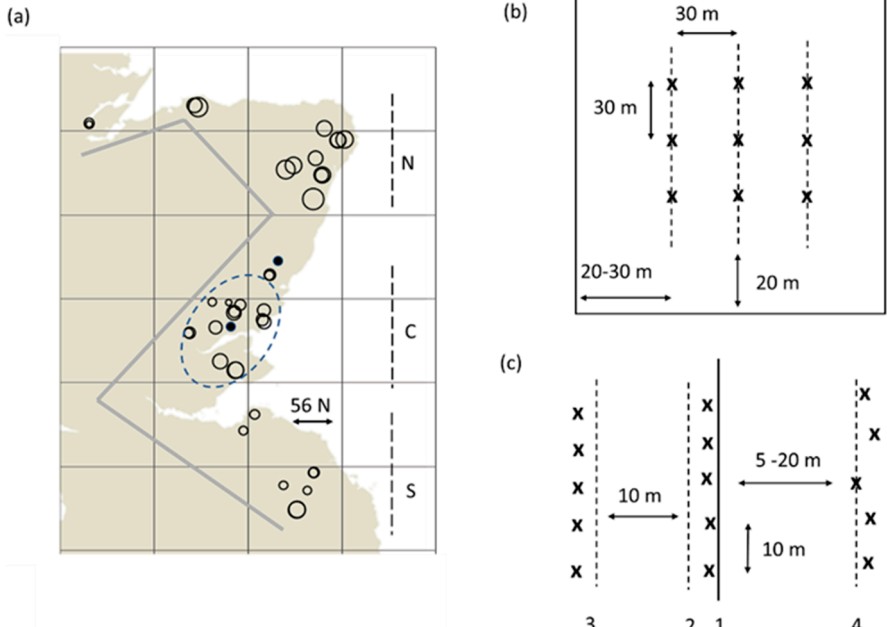

**Figure 1.** The study area and sampling plans: (**a**) map of sites, latitude 56N indicated, each site at the centre of an open circular symbol of size representing relative %C, the small closed symbols indicating locations of experiment stations used for context, the heavy zig-zag lines the approximate boundary of the oceanic climate in which arable crops and grass are cultivated, and the dashed oval the region in which fields, margins and uncultivated sites were sampled; (**b**) sampling locations (x) at each site along three transects, with typical dimensions indicated; and (c) indicative sampling plan in relation to field boundary (1) for comparison of field margin (2), field (3) and uncultivated land (4).

The present study concentrated on the high-input systems where previous work had found soils at risk, low SOC being associated with several other variables including high bulk density and low water holding capacity [22,23]. To facilitate analysis in this paper, fields were grouped in three intensity categories: A) highest input, equivalent to VI, B) winter and mixed cereals, V and IV, and C) spring cereals, occasionally with grass, mainly III. Thirty-eight fields were chosen for sampling in 2014 from an existing farm network [29]. Soils were mainly Stagnosols and Cambisols [28]. Criteria for selection included: fields had to be growing a crop at the time of sampling (rather than grass or fallow); as judged by background data, about half would be expected to fall in system VI (intensity group A) and half in systems V and IV (group B), with a few additional sites in systems of lower input; farmers or owners would be willing to share information on crops grown in the field over the past 12 years; and for consistency with previous studies [23,29], sites would be located across southern, central and northern sub-regions (shown in Figure 1a).

Soil samples were taken from late July to mid-August, before harvest of the crop, by a standard procedure [29] in which 9 samples were taken from three loci on each of three transects across each field (Figure 1b). The central transect was positioned close to the central point of a field boundary, then transects either side were located at a distance of at least 30 m. Other dimensions of the sampling scheme are shown in Figure 1b. In this region, the surface soils of arable fields have been typically well mixed by annual plough cultivation to a depth of 30-40 cm. At each sample locus, surface litter and vegetation were removed, then soil was mixed with a trowel and sampled to a depth of 20-25 cm, considered representative of soil to plough depth. Any large roots were removed and a sub-sample of the mixed soil was taken for analysis, dried at 70 °C, mixed further, and milled into a powder, of which 5 mg was processed for carbon and nitrogen concentration (%C and %N) by an Exeter Analytical CE440 Elemental Analyzer [29]. In this paper, SOC is used to refer in a general sense to soil organic carbon,

while %C and %N are used specifically to refer to outputs from the Elemental Analyser. The C:N ratio was obtained from measurements of %C and %N at each sample locus.

Sites were characterised for cropping intensity based on crop sequences over the period 2003–2014. As in a previous study [22], preliminary analysis showed SOC was correlated only with the proportion of potato and winter wheat in the cropping sequence for each field. Potato has the most intrusive tillage and most field traffic, by far the highest pesticide inputs of all crops (20–25 formulations annually), and high annual nitrogen and phosphate applications; while winter wheat has generally more traffic and more intrusive tillage than other major crops after potato, the second highest pesticide applications (10–12 formulations), and high annual nitrogen and phosphate application [22,30]. The analysis presented here, therefore, concentrates on the occurrence of these two crops. New to this study was the finding that about half those fields from high-intensity system VI had two potato crops in the last 12 years, while the rest had one. Therefore, the sampling in 2014 had defined a wider range of cropping intensity than previously, including fields growing up to 50–60% potato and winter wheat in the sequence and fields that did not grow either of these crops.

SOC at most high-input arable-grass sites is expected to be in a restricted range of 2–4%C [22,23,28]. Therefore, additional samples were taken to place the results within the broad context and full range of SOC in the region, and more specifically to set zones of SOC from high-intensity arable to uncultivated land. Soil from this additional sampling was processed by the same methodology and in the same laboratory as used to process the in-field samples referred to above. Soil was sampled in October 2015 in and near each of 10 sites within the area where fields of the highest intensity occur (Figure 1a). At each site, a field boundary was located, typically a stone wall or fence (labelled 1 in Figure 1c), then samples were taken from three 'habitats' in relation to the boundary: the uncultivated field margin (labelled 2 in Figure 1c); the tilled area of the field (3); and a nearby area of land that had not been under cultivation (4). Due to the varied shapes of the three habitats, the samples were not in all cases taken along parallel lines. The field margin was typically a strip of mostly perennial vegetation, 1-5 m wide, outside the tilled area but within the field boundary. The margins may have been cultivated or grazed at some time in the past. The uncultivated nearby land was located as close as possible to the field, generally 5–20 m distant, and comprised small areas of grass or woodland. It had probably never been cultivated but might have been lightly grazed in the past. An overall context for the data presented here is provided by national soil survey [25,27,28], which shows that under various forms of semi-natural vegetation, such as forest, unmanaged grass and wetland, SOC extends upwards to >40%. To ensure that the method of measuring %C in this paper is comparable, soil from outside the arable-grass was sampled at two of the James Hutton Institute's research farms, Balruddery and Glensaugh, identified in Figure 1a. Locations include (a) permanent grassland (4.4%C), upland deer grazing (12.3%C), and old woodland (rising to 41.7%C in a mainly organic soil).

## 2.2. Statistical Analysis

Differences in soil %C, %N and the C:N ratio across sites in terms of intensity groups A, B and C were analysed first by an Analysis of Variance for unbalanced designs using Genstat 18th edition (VSN International 2015), with intensity as a factor with three levels. Between-group differences were examined by post-hoc Bonferroni analysis. Variation in %C, %N and C:N ratio among sites by intensity (according to the proportion of winter wheat and potatoes in the rotation) was assessed by generalised linear regression, based on a normal distribution with an identity link function. Finally, the differences in the soil variables between habitats with different levels of disturbance were analysed with a generalised ANOVA on %C, %N and C:N ratio with habitat category declared as a factor with 3 levels (field, margin and uncultivated) and farm as a blocking term. Between-habitat differences were examined by post-hoc Bonferroni analysis.

## 3. Results

### 3.1. Comparison of SOC Over a Range of Cropping Intensity

Among the fields in Figure 1a, %C ranged from 1.60% to 4.64%, %N from 0.145% to 0.350%, and C:N ratio from 10.3 to 14.4 (Figure 2a). The range from low to high %C and %N occurred in each of the southern, central and northern parts of the range, with the exception that the three sites with highest %C were in the northern part. (The geographical ranges in Figures 1a and 2a are indicated to confirm that the location of sites in the present sample covered the same north-south extent as previous sampling in the study area [22,23] and are not examined further in this paper.) %C and %N both increased from intensity group A to B and C (F pr < 0.001). %C was 2.46% in group A, 3.17% in group B and 3.62% in group C (Figure 2b, differences between groups significant, P < 0.001). %N was 0.21% in group A, 0.26% in group B and 0.29% in group C (again, differences between groups significant, P < 0.001). C:N ratio was smaller at 11.77 in group A than in groups B (12.22) and C (12.63) (F pr. < 0.001) but did not differ between B and C. Table S1, Supplementary Material gives further output of the analysis.

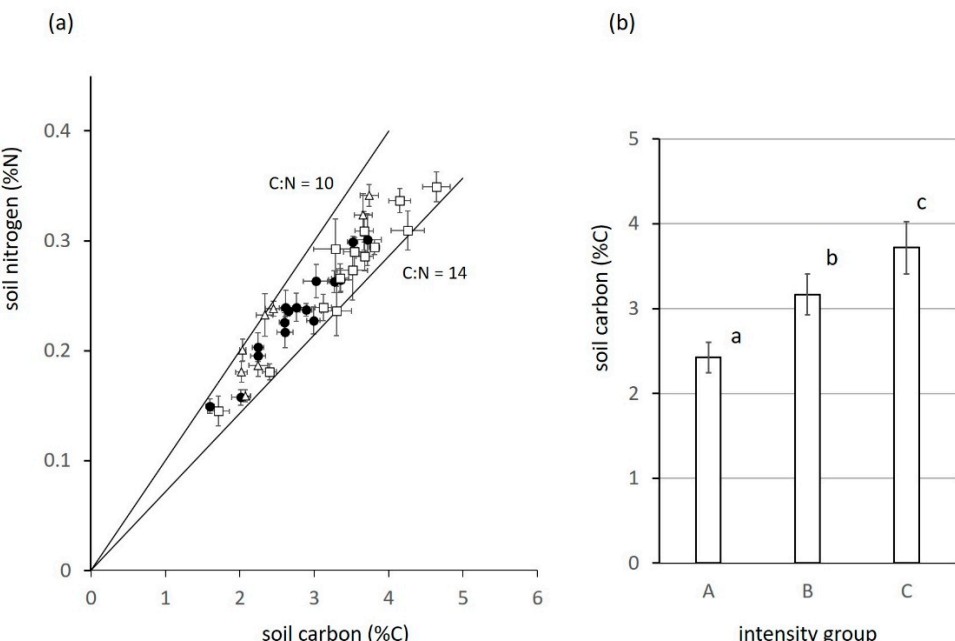

**Figure 2.** Summary of SOC data for main sampled sites showing (**a**) %C and %N (± standard error to show within-site variation) for northern (open squares), central (open triangles) southern (closed circles) regions, lines are for C:N ratios as indicated; (**b**) %C averaged over each of the three intensity groups, ± standard errors, effect of group significant (F pr. < 0.001), letters indicating significant differences between groups (A) potato and winter wheat in combination, (B) winter and mixed cereals, (C) mainly spring cereals and grass (text summarises statistical analysis).

The differences due to cropping intensity indicated above were refined through examination by site rather than group in terms of an intensity metric calculated as the proportion (p) of potato and winter wheat in the crop sequence (Materials and Methods). Both %C and %N showed a steep negative relation with rise in p from zero to 55% (Figure 3a,b). Generalised linear regression showed the trends were highly significant: %C = 4.02 − 0.038p, 59.7% variance accounted for, v.r. 47.0, F pr. < 0.001; and %N = 0.32 − 0.0028p, 55.6% variance accounted for, v.r. 41.5, F pr. < 0.001. Estimated reductions were from 4.02%C at zero p to 1.95%C at maximum p (decrease to 48% of value at zero p); and from 0.32%N at zero p to 0.17%N at maximum (decrease to 53%). In contrast, the regression of C:N ratio on p was not significant; its value was constrained around a mean of 12.1 (Figure 3c).

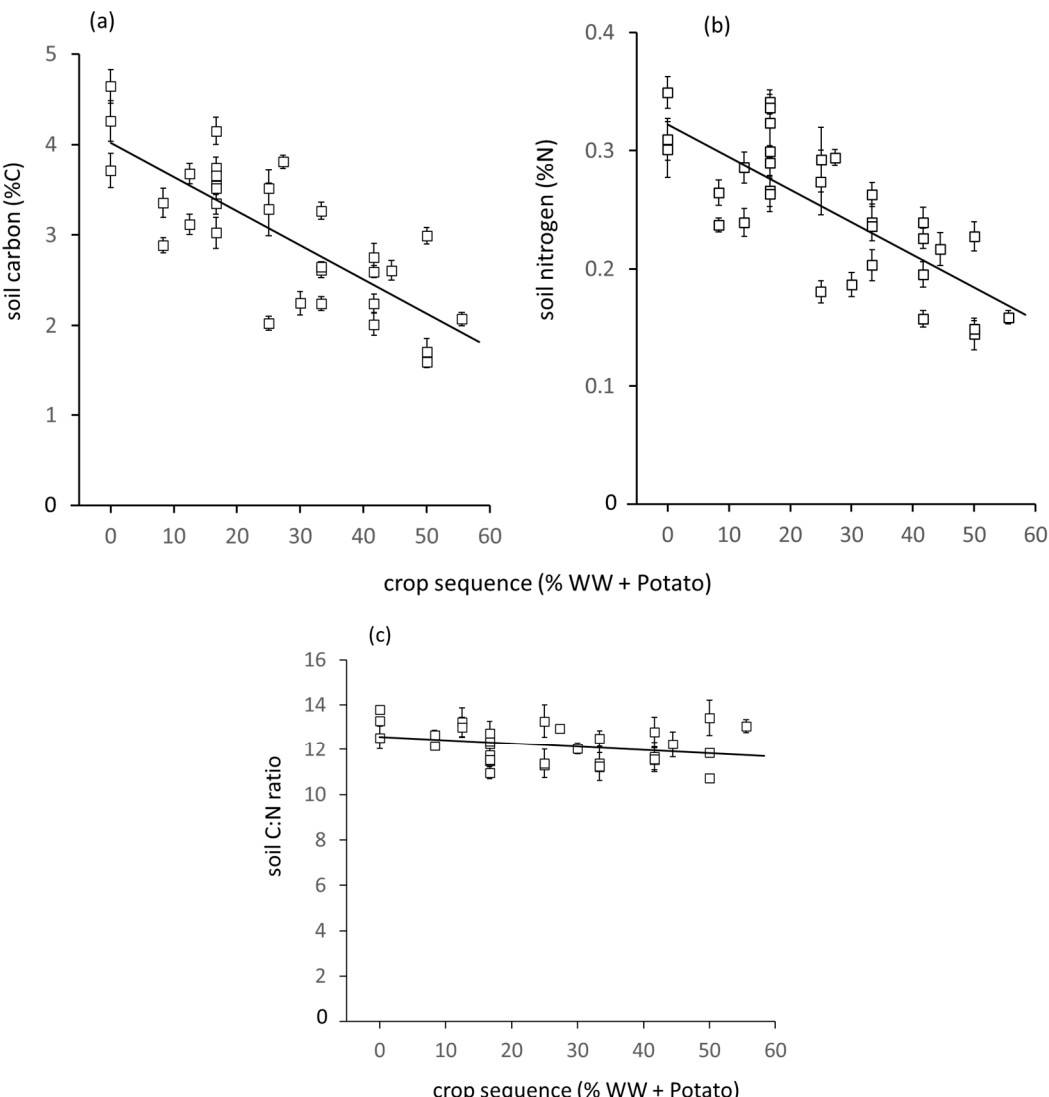

**Figure 3.** Variation in (**a**) %C, (**b**) %N and (**c**) C:N ratio in relation to the proportion of high-intensity crops, winter wheat and potato, in the crop sequence, symbols showing site mean ± standard error; regressions in (**a**) and (**b**) significant, F. pr. < 0.001 (see text), in (**c**) not significant.

### 3.2. Soil Carbon in Cultivated and Uncultivated Land

SOC was compared in the three habitats, field (F), field margin (M) and nearby uncultivated land (U) at 10 further sites (Materials and Methods). Overall, ranges were 1.17% to 9.53% for %C and 0.10% to 0.77% for %N (Figure 4a). C:N ratio again ranged from 10 to 14 (Figure 4a). At all sites, %C and %N were smaller in F than in M and U. Analysis of variance confirmed effects of habitat on all three variables. Mean %C (Figure 4b) was 2.39% at F, 3.34% at M and 4.79% at U (F pr. < 0.001, all between-habitat differences significant). Mean %N was 0.206% at F, 0.262% at M, and 0.383% at U (F pr. < 0.001, all between-habitat differences significant). Mean C:N ratio was 11.6 at F, 12.8 at M and 12.5 at U (F pr. < 0.001, F different from M and U, but M and U not different from each other). Statistical output is summarised in Table S1, Supplementary Material.

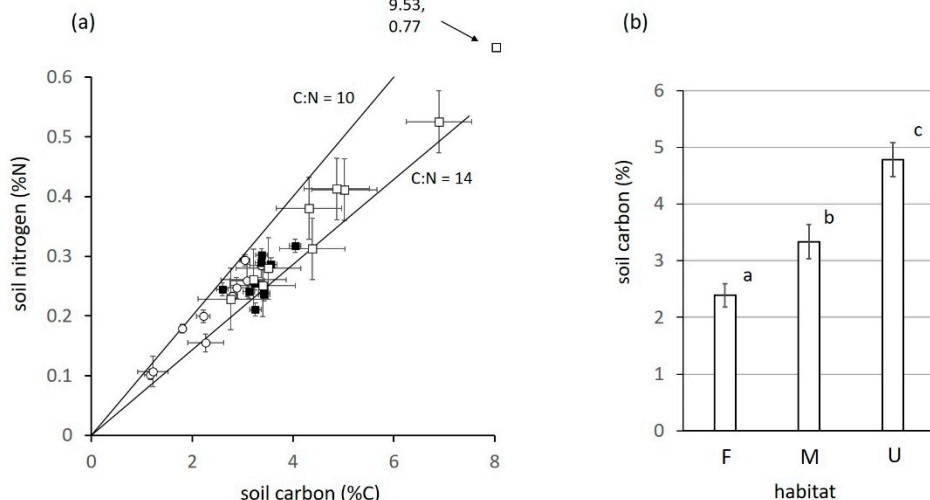

**Figure 4.** Variation in SOC among three habitats, field (F), field margin (M) and nearby uncultivated (U) areas: (**a**) %C against %N per site for F (open circles), M (closed squares) and U (open squares), ± standard errors to show within-site variation, lines showing C:N ratios as indicated, coordinates of an outlier site given; and (**b**) results of ANOVA showing effects of habitat on mean %C ± standard errors (F pr. < 0.001), letters above bars indicating significant differences between habitats, details in text.

## 4. Discussion

Different intensities of land management and the retention of semi-natural vegetation in this region have resulted in an extensive range in SOC among arable-grass fields and between fields, field margins and uncultivated land. The findings imply that the many farmland mosaics in Europe should not be treated as uniform when assessing the risk of SOC decline. The lowest SOC values, down to near 1% were found at the high-input end of arable cropping, characterised by very intrusive soil cultivation and the highest pesticide and fertiliser inputs. At the low-input end, where spring cereals were prevalent, sometimes with rotational grass, SOC was typically >3% and extended above 4%. Compared to this variation in SOC, the C:N ratio was, as expected [2] far more constrained, at an overall mean of 12 and range between 10 and 14. The strong effects on C:N of climatic differences found across some regions of the world [7] can be ruled out here. The slightly lower C:N ratio in fields than field margins was likely related to intrusive tillage over many years. The mechanisms responsible would be difficult to define without further analysis of soil microbial processes [5]. Given the present results, however, C:N ratio would not be a sensitive indicator of different degrees of farming intensity among cultivated fields.

The main finding is the strong negative relation between cropping intensity and both %C and %N in Figure 3a,b. The relation is likely to be causal, but absolute confirmation is lacking. There are few systematic records of SOC in arable-grass cropland before intensification in the mid-1900s [26]. The tilled land at that time consisted of mostly spring-sown arable and rotational ley, shallow ploughed with horse traction and receiving little fertiliser and very little pesticide [20]. New practices arose from the 1960s, including deep ploughing to 30-40 cm by heavy tractors, higher mineral fertiliser, and increasing pesticide. However, measures of SOC in arable land after intensification have not been consistent in demonstrating a trend, some studies recording a decrease, others no change [25,26,31]. A major lesson from the present study is that 'arable' is too broad a category for comparison between studies and time points. In particular, its use as a general term for a wide range of practices may obscure SOC declines in the most intensely managed fields. Nevertheless, if SOC measured in soils of low-input arable, pasture and field margins is indicative of states before intensification, then a change of -1%C over a time scale of 60 years would be broadly consistent with the losses cited for other temperate regions following intensification [1,4,12,31].

The full range in SOC in the region extends well above values found in arable soils [28]. For assessment and management, therefore, it is not practicable to consider a region such as this as a uniform class, which is the almost inevitable result when mapped based on broad categories of land use [16]. The region is more accurately described by one of the definitions of HNVF [17]: "mosaics of semi-natural and cultivated land and small-scale landscape features". While SOC over much of the area might be considered high in global comparisons of agricultural landscapes [1,4,26], attention should now be directed at fields severely at risk and in need of restoration.

*Soil Carbon Zones and Targets for Rehabilitation*

A complex relation exists between soil carbon and broad, soil-dependent functions such as primary production and yield. There is little evidence for a critical threshold (such as 2%) below which functions deteriorate [32]. Moreover, soil properties may depend on the proportions of particular fractions of soil carbon rather than on the total [33,34]. Nevertheless, a feasible, achievable target SOC should be definable for high-input fields. The lower range for arable fields in this study can be taken broadly as 1%C and the upper range as 5%C. Values higher than this have been found in other work: permanent pasture in this region and elsewhere in the Atlantic zone has been measured around and above 5%C [5]; while a separate study on arable-grass in part of the northern area found a quarter of sites between 5%C and 6%C and a maximum at 6.8%C [26].

Given that very few arable sites sampled here were approaching 5%C, a value somewhat lower may be more realistic when considering an initial target for reversing declines in SOC. To assist in the definition of this target, the range and interquartile are given in Figure 5 for several site categories. For those in the high-intensity group A, SOC was mostly below 3%C. Those sites of very high intensity, specifically having more than 10% potato in the crop sequence, were lower than 2.5%C (Figure 3a). Most field margins in the high-intensity region (Figure 1a) were (despite one high and one low outlier) in the range 3.2% to 3.4%, on average +0.96%C of the value in the immediately adjacent field. This raised SOC in field margins can be taken as a measure of the carbon and nitrogen that has accumulated or remained in the same soil and microclimate in the absence of intrusive cultivation. SOC above 3%C is also found at sites in the low-intensity group C. These arable-grass sites above 3%C are growing the same crops as those in groups of higher intensity, but simply with a lower frequency of the high-input crops. A target of >3%C or typically a change of around +1%C should therefore be attainable for sites having low SOC. The fields and other locations in the study can, therefore, be placed in three zones (Figure 3): Zone 1, from 1%C to >3%C (placed near 3.2%C in the diagram), the current range of high-input arable; Zone 2, from 3.2%C to 5%C for low-input arable and arable-grass; and over a much wider range, Zone 3, above 5%C, mostly occurring outside managed agriculture but including some pasture and arable-grass.

Two questions now need further examination: the extent of land at risk and options for rehabilitation. In order to understand the association between cropping intensity and SOC, this study purposely comprised a greater proportion of high-input fields than occurs on average in the region. The trends in Figure 3 are, therefore, open to the sort of bias found in erosion studies that have targeted at-risk sites rather than selecting an unbiased sample [24]. The fields prone to loss of SOC (in the lower half of the range in Figure 3a,b) would be primarily from high-input systems VI and V. From analysis of data from the EU's Integrated Administration and Control System or IACS [35], these systems occupied between them 17% of the arable-grass surface, while the system of next highest intensity (system IV, mixed cereals) occupied a further 21% [21]. In principle, therefore, with knowledge of the history of each field, it should be possible to define specific fields at high risk within these cropping systems.

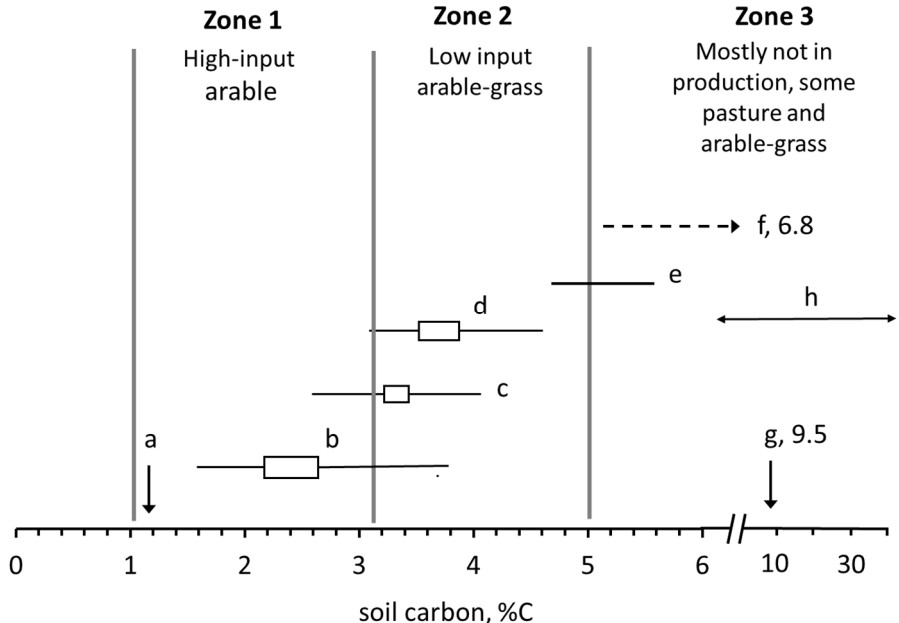

**Figure 5.** Division of soil organic carbon into three zones: (1) mainly high-input arable, (2) low-input arable-grass, and (3) land mostly outside agricultural production (e.g., woodland, bog, unmanaged grass) but including some pasture and arable-grass. Indicative SOC values and ranges are shown by letters: (a) minimum site mean recorded in this study, 1.17%, (b) high-input range and interquartile, (c) field margin range and interquartile, (d) low-input range and interquartile, (e) range in a pasture experiment [5], (f) upper limit for arable grass in [26], (g) maximum in uncultivated land close to fields (this study), and (h) range for uncultivated or semi-natural locations.

The second and more challenging question concerns the method of rehabilitation and its consequences. SOC consists of many fractions which accumulate and degrade at different rates [34,36,37]. Tillage can transfer SOC from stable to active pools with residence times of only a few weeks [34], while organic carbon from the structural material of roots is less prone to breakdown and loss than are root exudates [38]. Soil microbes compete for and release organic matter at rates depending on the stoichiometry of C with N and P, which may depend on long-term fertiliser regimes [5]. Despite such complexity, experience has shown that soil SOC can be increased in arable fields. Both experience and modelling agree that the conversion of arable to pasture would give the greatest rate of increase in C storage, but other interventions such as short-term leys and cover crops, particularly legume forages [37], could also be effective [1,11,12]. Some practices such as reduced tillage, while raising SOC in some situations [39] may in others alter the depth-distribution of SOC with little increase in the total [40]. Even applying some or all of these changes, the rate of accumulation is likely to decrease year on year before settling at a new equilibrium [41], which might be lower than that before the original decline due to intensification [1]. Caution should be exercised however when estimating gain in soil carbon stocks from rehabilitation. Assessment of the capacity of land to sequester carbon needs to account for the full range of carbon stores in soil and vegetation, including possible change in depth of soil horizons due to intensification [25].

Restoration in this region would also need to accommodate the position that agriculture here is, and has been for thousands of years, geared to a varied crop and grass output [20], among which present high-input systems are the most profitable [42]. Such a wide range of crops and livestock also provides resilience against economic and climatic shocks. Conversion of most arable to long-term pasture could, therefore, be detrimental to the viability of agriculture and food security. A conversion to rotational grass in some fields coupled with a more integrated approach to arable management [43] may offer a practical solution. Models predicting rates of gain or loss in relation to the type of carbon inputs could be attuned to the local climate and soil to guide the choice of options [12]. Crucially,

management and restoration have to be implemented at the scale of the management unit (field) rather than by treating the region or even a small part if it as a uniform block. Given the wide range and combination of crops grown, the fields most at risk can only be detected from knowledge of decadal crop sequences.

## 5. Conclusions

The diverse, Atlantic zone cropland studied here contained a wide range of variation in SOC, both among fields and between fields and their immediate surrounds. Agricultural landscapes such as this should not be treated as uniform, whether in assessment and restoration of SOC or estimate of carbon sink. Crucially, the occurrence of high-input crops grown over more than the previous decade in each field was needed to quantify a steep negative association between SOC and cropping intensity, while the higher SOC in uncultivated field margins and nearby land offered a base from which to assess in-field decline and a realistic target for rehabilitation. The widespread practice of grouping land use into broad terms such as 'arable', 'grassland' or HNVF and non-HNVF for assessment of SOC status and decline is, therefore, likely to be misleading. Field by field assessment of the risk to SOC and the need for rehabilitation is now feasible in Europe, based on the incidence of high-intensity crops detected using IACS data [21,25]. Further definition is needed of the best means of raising SOC while maintaining a varied agricultural output. Understanding SOC status and potential will be aided if precise causes of change were identified in terms of shifts in SOC components and microbial activity following intensification and during rehabilitation. Methods for raising SOC need to be tested experimentally in these landscapes [43], since experience in other studies suggest no single approach is likely to work [1,12,39].

**Supplementary Materials:** The following are available online at http://www.mdpi.com/2073-4395/10/7/973/s1, Table S1: Analysis of Variance results for (a) the effect of intensity groups A, B and C, and (b) the effect of habitat (field, margin and uncultivated land), on %C, %N and the C:N ratio.

**Author Contributions:** Conceptualization, G.R.S. and C.H.; methodology, G.B. and L.F.; data curation, G.B., L.F. and M.Y.; formal analysis, C.H., L.F., M.Y., G.R.S.; validation, M.Y. and G.R.S.; writing, G.R.S. and C.H. All authors have read and agreed to the published version of the manuscript.

**Funding:** This work is supported by the Strategic Research Programme of the Scottish Government's Rural & Environmental Science and Analytical Services Division (RESAS).

**Conflicts of Interest:** The authors declare no conflict of interest. The founding sponsors had no role in the design of the study; in the collection, analyses, or interpretation of data; in the writing of the manuscript, and in the decision to publish the results.

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
