# Peer review of "Defining Targets for Reversing Declines of Soil Carbon in High-Intensity Arable Cropping"

_agronomy, doi:10.3390/agronomy10070973_

Round 1
Reviewer 1 Report
I would like to thank the authors for providing a thorough and detailed review response. The authors have addressed all my comments.
Good job in the revision.
Reviewer 2 Report
An interesting article in a good presentation.
This manuscript is a resubmission of an earlier submission. The following is a list of the peer review reports and author responses from that submission.
Round 1
Reviewer 1 Report
In the study, which is based on Scotland’s arable grass cropland, the authors have sampled commercial cropland fields to define the causes of SOC decline and setting restoration targets for SOC. The results of the study showed that among all cropland types- high and mid-intensity sites had SOC variability mostly due to cropping intensity and lesser effect of latitude. Overall, this is an interesting study providing insights into mosaics of farmlands and supplementing broad agricultural land classification. The outcomes would be helpful for managers and farmers alike in restoring farmlands.
However, I found several concerns on the paper – largely of statistical analyses and results presented, organization, formatting, and styles.
Specific comments:
Title: The paper is about SOC, so instead of “Soil Carbon”, “soil organic carbon” may better suit
Abstract:
Line 17: if the p-value is significant, how the effect is lesser?
Line 19: what is the p-value?
Introduction:
Line 30: Please mention which greenhouse gases you are referring to here. Writing just greenhouse gases is a vague statement.
Lines 26, 34, 53: Please define abbreviations when writing for the first time i.e. EU, N, P, etc. and that applies throughout the manuscript.
Line 49-50: Needs references
Line 87: Remove space
Material and methods:
ANOVAs and generalized regressions: need an explanation why specific ANOVA tests were performed and what uncertainty existed? Also, readers would follow the analyses easily if equations were provided! Another concern is that the authors did not mention how the averages were compared after ANOVA or regressions.
I suggest the authors include the study site map in this section rather than in the result section so that readers can get a clear idea where the sampling was done in Scotland and how the study was designed. At first, I thought the figure was entirely missing from the paper!
Lines 98-102: Schematic diagrams or detailed site pictures showing sample plots and transects would be helpful.
Line 99: How widely spaced?
Line 100: Why 20-25 cm depth? Needs explanation and reference.
Line 102: Scientific instruments/tools should be cited properly.
Line 119: I think, authors here meant to say “experiments” not experience.
Lines 114-116: As mentioned below please include all statistical information in a separate subsection for the clarity of the text and provide equations and explanations.
Line 123: Please watch the formatting of the sentence!
Lines 124-126: What was the depth of soil samples collected and how the sampling was done? Please describe that in detail in the text! Also, mention in which month of the year the sampling was done?
Lines 139-148: Write a separate paragraph for the statistical information.
Line 141: Software should be cited properly.
Results
Results related to the ‘target’ setting for the restoration should be presented in this section than entirely writing in the discussion section. e.g. Lines: 243 -263
I strongly recommend including model outcomes in tables so that readers can see all the parameter values.
Throughout the results, it is not clear how group averages were compared e.g. lines 173-175 where C:N ratio for northern (12.64) and southern (11.87) are reported, but it is not clear if those average values were calculated separately or are resulted from posthoc comparisons! Also, in all of the group comparisons, t-values should also be reported along with p values.
Moreover, rather than stating as variance ratio or “v.r.”, F-values with degrees of freedoms in parentheses may be a common practice – though it could be a disciplinary format!
Lines 158-181: Either write variance ratio or VR. Make it uniform throughout the manuscript!
Lines 178-180: Genstat is a software, not a test!
Lines 183-188: Should be in the method section.
Discussion
Soil carbon dynamics is a complex process and in this paper, at least in the discussion, authors should acknowledge that there are still uncertainties in the outcomes as they did not account for the role of soil moisture content, soil microbial decomposition, and sources and loss of organic carbon (e.g. parent material, drainage, DOC, POC), etc.
The result portion of the soil target zoning (subsection: “soil carbon zones and rehabilitation”) should be reported in the results and discussed here.
Concluding part of the paper or author’s concluding remarks based on the study, explanation of uncertainties related to smaller sample numbers and studied limited factors for carbon loss, and accordingly, the recommendation for future studies building on this study and/or overcoming uncertainties should be provided.
Line 203-204: “….. the finding……decline.” should be in the conclusion.
Line 217: “C:N” should be “C:N ratio” throughout the paper.
Line 231: Reference missing.
Line 239-240: which study? Reference required!
Line 240: Rather than saying ‘a few’ please be specific with the numbers.
Line 232: Indent is missing from the starting of the paragraph.
Line 247: Watch sentence! Proofreading required for all sentence structure and grammar.
Line 264 onwards: font and formatting changed!
Line 264 – 280: This paragraph has no or very weak link with the previous paragraphs. I suggest, rephrasing the paragraphs for better writing.
Reviewer 2 Report
I have practically no comments. I think the article will benefit if the authors make a few clarifications.
- How samples were prepared for elemental analysis (removing of roots and parts of plants, sample dispersion...).
- Give in the article soil names according to WRB classification